# Antioxidant, Anti-Diabetic, and Anti-Inflammation Activity of *Garcinia livingstonei* Aqueous Leaf Extract: A Preliminary Study

**DOI:** 10.3390/ijms25063184

**Published:** 2024-03-10

**Authors:** Murendeni Nethengwe, Nasifu Kerebba, Kunle Okaiyeto, Chinyerum S. Opuwari, Oluwafemi O. Oguntibeju

**Affiliations:** 1Phytomedicine and Phytochemistry Group, Department of Biomedical Sciences, Faculty of Health and Wellness Sciences, Cape Peninsula University of Technology, Bellville 7535, South Africa; 219477221@mycput.ac.za (M.N.); okaiyetok@cput.ac.za (K.O.); 2Department of Chemistry, Makerere University, Kampala 7062, Uganda; nkerebba@gmail.com; 3Department of Medical Biosciences, University of the Western Cape, Bellville 7535, South Africa; copuwari@uwc.ac.za

**Keywords:** antioxidants, diabetes mellitus, *Garcinia livingstonei*, hyperglycaemia, inflammation, oxidative stress, phenolic compounds

## Abstract

Diabetes mellitus (DM) is the second leading cause of mortality globally. The increased concern for DM is due to the underlying complications accompanying hyperglycaemia, associated with oxidative stress and consequent inflammation. The investigation of safe and effective treatments for DM is necessary. In the present study, the cytotoxicity, phytochemical analysis, antioxidant capacity, anti-inflammatory, and antidiabetic effects in an aqueous extract of *Garcinia livingstonei* leaves were assessed. All tested extract concentrations showed no toxicity against C3A hepatocytes. Several phenolic compounds were identified using ultra-high performance liquid chromatography mass spectrometry (UHPLC-MS). The total polyphenol content was 100.9741 mg GAE/g, 16.7712 mg CE/g flavanols, and 2.3548 mg QE/g flavonols. The antioxidant capacity values were 253.4268 mg AAE/g, 192.232 mg TE/g, and 167.8724 mg TE/g for ferric reducing antioxidant power (FRAP), Trolox equivalent antioxidant capacity (TEAC), and 2,2-diphenyl-1-pycrylhydrazyl (DPPH), respectively. The plant extract significantly (*p* < 0.05) demonstrated anti-inflammatory and hypoglycaemic effects in a dose-dependent manner, with the α-glucosidase inhibition of the extract being higher (*p* < 0.05) than in the standard conventional drug (acarbose). The findings of this study revealed the potential of the constituents of *G. livingstonei* aqueous leaf extract in DM treatment. Further studies on the preparation and mechanisms of action of the plant in DM treatment are recommended.

## 1. Introduction

Diabetes mellitus (DM) is a chronic, non-communicable metabolic disease distinguished by the emergence of hyperglycaemia [1]. Alteration in the production or secretion, and the signal transduction of insulin in DM patients is the main cause of hyperglycaemia [2]. DM is categorized into two major types, namely, type 1 DM, caused by β-cell destruction, and type 2 DM, characterized by insulin resistance [3]. The pathogenesis of DM is associated with oxidative stress and inflammation, and both are implicated in hyperglycaemia [4]. Excess glucose molecules lead to oxidative stress through mitochondrial dysfunction and the glycation of biomolecules subsequently leading to chronic inflammation [5,6]. Oxidative stress occurs due to the dysregulation of free radicals caused by an imbalance between free radical formation and elimination by the antioxidant system [4]. Excessive free radicals interact with biomolecules and consequently lead to tissue damage and chronic inflammation [7]. Chronic inflammation occurs when the activation of immune cells and the secretion of inflammatory cytokines exceed normal levels [8]. The infiltration of immune cells and direct accumulation of inflammatory cytokines in organs lead to damage and further pathogenesis of DM [9]. The implications of hyperglycaemia explain the accompanying complications of DM, such as cardiovascular diseases, neurological and neurogenerative diseases, nephrological damage, and male infertility [10,11,12].

DM is a global public health problem with a prevalence of over 546.6 million people diagnosed in 2021 [13]. It is estimated that approximately 35.1 billion US dollars will be spent globally by 2030 towards the treatment plan, hence, the need for effective and affordable treatment [14,15]. Reduction of hyperglycaemia and alleviation of oxidative stress and inflammation is crucial in the treatment of DM-complications [16]. The inhibition of carbohydrate metabolism enzymes (α-glucosidase and α-amylase) reduces hyperglycaemia [10]. In addition, several studies have uncovered the role of lipid breakdown and absorption in the manifestation of obesity and have discovered the effect of lipase inhibitors in the treatment of obesity and DM [17]. Conventional treatment of DM includes metformin [2], as a first line of treatment, and other drugs such as acarbose, voglibose, and glibenclamide [18]. However, these synthetic drugs exhibit adverse effects, are highly expensive in production, and unaffordable in most developing countries, such as South Africa [2]. It is therefore paramount to investigate potential effective, safe, and affordable therapies for DM. The literature reveals some medicinal plants that are used for the treatment of DM with an interest in their phytochemical content, such as antioxidants and anti-inflammatory compounds [1,19]. However, many of the studies have only identified the plants (ethnobotanical studies), and limited studies have been done on the phytochemical profiling and evaluation of the antidiabetic effects that these plants possess [20].

*Garcinia livingstonei* is an evergreen plant indigenously found in Africa and popularly known in South Africa for its beneficial use [21]. In South Africa, the plant is found in some areas of Limpopo, where it is known as “Mupimbi” in the Tshivenda language. *G. livingstonei* bears green leather-like leaves and yellowish orange round edible fruits [22]. In different countries in Africa, *G. livingstonei* is identified by traditional healers and herbalists and used for the treatment of ailments such as bacterial infections, infertility, diabetes mellitus, and some respiratory complications through the preparation of either its leaves, stem, or roots [22]. Although only a few studies have been conducted on the leaves of the plant, review studies have revealed the possibility of the leaves of the plant having therapeutic effects in the treatment of DM due to their constituents [23]. Several phytochemicals of interest, such as benzophenones, phenolic acids, and flavonoids, have been profiled, leading to reports of the antioxidant, antibacterial, and antiviral properties of the plant [21,24]. However, information and data revealing the antidiabetic effect of the leaves of *G. livingstonei* are insufficient. From the current limited information in the literature, our study focused on the possible effect of the leaves of *G. livingstonei* in the treatment of diabetes mellitus by investigating the effect of the aqueous extract on the major carbohydrate enzymes, identifying possible responsible phytochemicals in the extract, and determining the antioxidant and anti-inflammatory capacity of the extract.

## 2. Results and Discussion

### 2.1. Cytotoxicity Effect of Aqueous G. livingstonei Extract

The toxicity effect of the aqueous extract of *G. livingstonei* was evaluated on C3A hepatocytes to determine what concentrations of the extract are safe to use for treatment in subsequent experiments, as depicted in Figure 1. The cell viability of C3A hepatocytes treated with all different concentrations (15.625, 31.25, 62.5, 125, 200, and 250 μg/mL) of the plant extract was significantly (*p* < 0.05) higher compared to the negative control cells treated with melphalan (including the concentration used). There was an observed significant (*p* < 0.05) increase in cell viability in the first four lower concentrations of the aqueous extract (15.625, 31.25, 62.5, and 125 μg/mL) of *G. livingstonei* compared to the negative control (untreated C3A hepatocytes), suggesting the effect of the lower concentrations of the extract in promoting cell viability. The two highest concentrations of the extract (200 and 250 μg/mL) exhibited significantly (*p* < 0.05) lower cell viability compared to the lower concentrations, although there is no significant (*p* > 0.05) difference between the positive control and these two highest concentrations. It is evident that higher concentrations than the ones tested in the present study could possibly exhibit minimal toxicity effects, considering the decrease in cell viability as the concentration of the extract increases. There was no significant (*p* > 0.05) decrease in cell viability of the C3A hepatocytes treated with all concentrations of the extract tested, compared to the untreated control C3A hepatocytes. These results reveal the safety of the use of this plant extract and the necessity of studying higher concentrations for possible higher efficacy.

### 2.2. Quantification of Specific Phenolic Compounds

The quantification of the present bioactive compounds of the aqueous leaf extract of *G. livingstonei* revealed the composition of bioactive chemicals (polyphenols) previously reported to be agents of therapy in the treatment of diabetic complications [23]. In previous studies, isolated polyphenols were used in drug therapy, to produce less-toxic drugs to treat diabetic complications. The current study shows a high level of total polyphenols, comparable to other medicinal plants that were previously studied and found to have a free radical scavenging effect. Polyphenols serve a regulatory role in the balancing of the level of inflammatory cytokines in DM by reducing the expression of mRNAs coding for chemokines and cytokines [25]. Figure 2 shows the results of the quantification of the phenolic compounds. Amongst the specific phenolic compounds quantified in the study, flavanols (16.7712 mg CE/g) appear to be the highest of the total polyphenols (100.9741 mg GAE/g) in the aqueous extract of *G. livingstonei*, supporting their appearance as the vastest category of flavonoids [25]. No alkaloids were detected. The findings reveal the content of flavonoids as a possible contribution to the effects of the extract in the treatment of DM. The presence and predominance of flavonoids are possibly the cause of the high antioxidant capacity of the extract, as high levels of flavonoids have been linked to high amounts of TEAC in previous studies [26]. Flavonoids have also been reported for their hypoglycaemic effect, which can be linked with the inhibitory effect of the extract on carbohydrate metabolic enzymes [27].

### 2.3. Total Antioxidant Capacity

Hyperglycaemia-induced overproduction of reactive oxidative species leads to the oxidation of biomolecules, subsequently leading to DM complications [7]. The excessive production of free radicals effectuates the reduction of antioxidant capacity and subsequently exacerbates oxidative stress damage in DM [28]. The amelioration of antioxidant capacity is therefore a therapeutic target in the treatment of DM complications facilitated by oxidative stress. In this current study, the assessment of the antioxidant capacity of *G. livingstonei* aqueous leaf extract was carried out to evaluate its potential in the reduction of oxidative stress through the augmentation of the antioxidant system.

The assessment of the antioxidant capacity of extracts is also an alternative way of determining the appearance of antioxidant compounds. From the phenolic content findings, it is evident that the extract possibly has high reducing power and scavenging activity. These findings (Figure 3) are well supported by the high antioxidant capacity (DPPH = 167.8724 mg TE/g, TEAC = 192.232 mg TE/g, FRAP = 253.4268 mg AAE/g) exhibited by the plant in this study, with FRAP showing the highest value (253.4268 mg AAE/g). The ferric reducing power of the extract, shown by the FRAP value, reveals the ability of the plant to reduce free radicals by the donation of an electron, leading to the neutralisation of the free radical effect. Since the complications of DM are associated with the oxidation of tissues by reactive oxidative species, the observed antioxidant capacity values of *G. livingstonei* reveal its antioxidant and subsequent antidiabetic potential. The alleviation of oxidative stress in DM can relieve organ damage and increase quality of life in diabetic patients. An increase in oxidative stress consequently leads to chronic inflammation [6]. The potential of *G. livingstonei* in reducing oxidative stress leads to a downstream reduction of diabetic complications caused by inflammatory damage to physiological systems. Rising interest in the use of medicinal plants is due to the presence of phenolic compounds linked to an increase in antioxidant capacity [19]. Phenolic compounds such as hydroxycinnamic acids have been found to reduce free radicals such as superoxide anions through an increase in antioxidant enzyme activity, ultimately leading to an increase in antioxidant capacity values as observed in the present study [29]. Treatment with the leaf extract of *G. livingstonei* can possibly augment the antioxidant system in DM patients with compromised antioxidant enzymes and reduced non-enzymatic antioxidant proteins by directly reducing/neutralising free radicals. The antioxidant effect of *G. livingstonei* can potentially prevent further development of the pathogenesis of DM by halting the manifestation of oxidative stress-induced organ damage. In the pathogenesis of DM, oxidative stress is an instigator of insulin resistance caused by the impairment of the insulin signal cascade, and β-cell destruction [30]. The antioxidant effect of *G. livingstonei* leaf extract can contribute to the reduction of hyperglycaemia by increasing insulin sensitivity and relieving β-cell damage through the elimination of free radicals.

### 2.4. UHPLC-MS Identification of Phenolic Compounds 

Further phytochemical analysis by the using UHPLC-MS identified the presence of a total of 37 phytochemical compounds found in the aqueous leaf extract of *G. livingstonei*. The phytochemical analysis in the current study demonstrates that the aqueous leaf extract of *G. livingstonei* contains flavonoids such as flavan-3-nols (catechin, epicatechin, gallocatechin), flavones (apigenin-C-pentoside-C-hexoside) and benzophenones (Figure 4), which are highly active and can react with free radicals due to their free hydroxyl and carbonyl groups, which agrees with previous similar studies [31,32]. This section lists and explains the findings after the HPLC analysis of the extract.

#### 2.4.1. Identification of Flavan-3-Ols, Flavonols, and Flavone

As a supplement of the flavonoids detected in the quantification method, flavan-3-ols, flavonols, and flavone were detected using the UHPLC-MS analysis, and the compounds were detected from the chromatogram peaks in Figure 4. Flavan-3-ols in oligomeric/ polymerised forms (peaks 18–21) were identified as catechin and epicatechin. The monomers catechin, epicatechin, gallocatechin, and gallo(epi)catechin gave various lengths of polymers called proanthocyanidins (Pas). Flavanols are a group of flavonoids with three joined rings (pyron ring, catechol, and resorcinol) with hydroxyl groups, mainly responsible for the capacity of flavanols to eliminate free radicals [33]. The structural form of flavanols (epicatechin and catechin) determines the therapeutic role (antioxidative) they serve in the treatment of DM, depending on their polymerization and the position of hydroxyl groups [33]. Using the elution order and conjugation with deoxyhexose and hexose sugars, peaks 11, 20, and 21 could be tentatively identified. Additionally, in conjugation with galloyl (152 Da), peak 23 could be tentatively identified as (−)-epicatechin-3-Ogallate. The methylated sulphate derivative peaks 30 and 31 could also be identified with corresponding stereoisomers. Oligomeric forms were identified according to previous literature in peaks 13, 24, and 25 [34].

Peaks 12, 14, 22, 28, and 32 were identified as flavanol peaks after conjugation of aglycones with various sugars (hexose, 162 Da, and pentose, 146 Da). The aglycone isorhamnetin could appear in a free form at peak 28. The flavone, apigenin-C-pentoside-C-hexoside was identified in line with previous reports with the aid of an accurate mass match, *m*/*z* 563.2272. The hypoglycaemic effect of flavonoids is linked to the modulation of the insulin signalling pathway during glucose metabolism through the activation of Protein kinase B (Akt) and insulin receptor-1 (IRS-1) enzymes to increase glucose uptake by cells [35]. In DM, the oxidative stress-induced apoptosis of cardiomyocytes can be alleviated by the elimination of free radicals [36]. Flavones have been previously documented to exhibit cardio-protective effects in DM by reducing free radicals through both an increase in antioxidant enzyme activity and inflammatory cytokines [37]. Therefore, presence of flavonoids (flavanols, flavones, and flavonols) in the aqueous leaf of *G. livingstonei* contributes to the antidiabetic potential of the plant as it can possibly ameliorate hyperglycaemia, oxidative stress, and inflammation.

#### 2.4.2. Identification of Benzophenones

In Figure 4, Peaks 7 and 8 were identified to belong to benzophenones, with the aglycone ion at [M−H]^−^; *m*/*z* 272. The fragmentation of the compound in peak 8 is similar to the one found in the literature for 3^1^-β-Glucosyloxy-4,4^1^-dihydroxy-2,6-dimethoxy-benzophenone [38]. Benzophenones are organic phenolic compounds with two interconnected C-rings as their basic structure. Various benzophenone compounds, artificial or naturally occurring, are derived from benzophenones to elicit therapeutic effects depending on the moiety and azines in the compound [18]. The benzophenone identified in this study (3^1^–β-Glucosyloxy-4,4^1^-dihydroxy-2,6-dimethoxy-benzophenone) was previously detected in a study that specifically profiled benzophenone derivatives in *G. livingstonei* [38]. Glycoside derivatives of benzophenones were previously reported to exhibit α-glucosidase inhibitory activity [18]. The appearance of 3^1^–β-Glucosyloxy-4,4^1^-dihydroxy-2,6-dimethoxy-benzophenone as a constituent of *G. livingstonei* leaf extract suggests possible antidiabetic effects through the delay in starch breakdown, which subsequently reduces blood glucose concentration. Most of the benzophenones reported from the genus *Garcinia* are polyisoprenylated benzophenones, derived from maclurin, a phenolic compound with ROS scavenging capacity [39]. The literature has also reported the effect of benzophenones in the management of obesity as a preventative measure against the development of DM. Benzophenones improve dyslipidaemia which leads to a reduction in body fat and body weight [40]. It was also reported that benzophenones promote fuel utilisation through carbohydrate metabolism by promoting glucose oxidation [40]. The shift of fuel utilisation to carbohydrates can ultimately lead to a reduction in blood glucose concentration, i.e., the amelioration of hyperglycaemia. Besides the hypoglycaemic effect, the anti-inflammatory effect of benzophenones was revealed through the reduction of major cytokines such as TNF-α and IL-1β observed with the treatment of cells with benzophenones [41].

#### 2.4.3. Identification of Hydroxycinnamic Acids

The UHPLC-MS analysis (Table 1 and Figure 4) also reveals the presence of hydroxycinnamic acids (dimethoxycinnamic acid, vanillic acid hexoside, p-coumaric acid, and 3-O-malonyl-5-O-€-caffeoylquinic acid). Peaks 1, 3, 4, 15, and 16 could be identified as those of hydroxycinnamic acids. Peak 3 was tentatively identified as p-Coumaric acid ethyl ester due to the presence of *m*/*z* 119, which would represent the decarboxylated coumaroyl. A free dimethoxycinnamic acid was also assigned at peak 1. Vanilic acid hexoside could be assigned on peak 4 due to the vanilic acid ion [M−H]^−^; *m/z* 167, which indicated loss of hexoside from the precursor ion *m*/*z* 329.0814. Acylation with malonyl (86 Da) on caffeoyl quinic acid led to the compound in peak 16 being tentatively identified as 3-O-malonyl-5-O-(E)-caffeoylquinic acid and its derivative at peak 15. 3-O-malonyl-5-O-(E)-caffeoylquinic acid was first identified in the fruits of wild eggplant [42]. P-coumaric acid is a natural phenolic compound derived from cinnamic acid after a shift in the position of the hydroxyl group. The antioxidant effect of *G. livingstonei* can be associated with the presence of p-coumaric acid, previously reported to increase DPPH and TEAC values, and reduce superoxide and hydrogen peroxide (free radicals) [43]. In a previous study, 3-O-malonyl-5-O-€-caffeoylquinic acid was isolated from *Solanum incanum* and found to increase antioxidant capacity through an increase in the DPPH value [44]. Dimethoxycinnamic acid alleviates lipid peroxidation, inflammation, and hyperlipidaemia, i.e., it also contributes to the therapeutic effects of *G. livingstonei* [45]. Gallic acid, too, has previously been found to exhibit antioxidant potential. Hydroxycinnamic acids are well-documented for their anti-inflammatory effect, hence the anti-inflammatory effect observed [29]. The anti-inflammatory activity of hydroxycinnamic acids is achieved by the reduction of pro-inflammatory cytokines and the blockage of macrophage infiltration [46]. These compounds also promote an increase in adiponectin secretion in the reaction against inflammation [46]. It has also been recorded that hydroxycinnamic acids are free radical scavengers targeting the eradication of superoxide, reducing lipid peroxidation, and increasing antioxidant capacity by increasing antioxidant enzyme activity [29].

### 2.5. Anti-Inflammatory Activity

The advancement of DM is linked to the excessive release of pro-inflammatory markers in reaction to oxidative stress damage [67]. Over-activation of the inflammatory response leads to more organ damage, more production of free radicals, and the development of DM complications [68]. Therefore, anti-inflammatory agents are of great importance in the treatment of DM. To complement the findings that the aqueous extract of *G. livingstonei* contains anti-inflammatory compounds, anti-inflammatory activity was determined and indicated by the decrease in nitrite concentration in response to LPS activation of RAW macrophages with no effect on cell viability, as seen with the AG control-treated cells (Figure 5A). Although treatment with *G. livingstonei* aqueous extract showed a significant (*p* < 0.05) decrease in cell viability (as depicted in Figure 5B), all tested concentrations (50, 100, and 200 μg/mL) of the extract, maintained cell viability over 100%, which validates the observation of nitrite production in the determination of anti-inflammatory activity. A significant (*p* < 0.05) decrease in nitrite concentration in a dose-dependent manner is observed when compared to the lipopolysaccharide (LPS)-activated cells, with the highest dose (200 μg/mL) showing a significantly (*p* < 0.05) lower concentration of nitrite compared to the other concentrations of the extract (Figure 5A). Although the treatment of the cells with aminoguanidine (AG), as a positive control, showed a higher level of anti-inflammatory activity than the plant extract, significant (*p* < 0.05) and commendable anti-inflammatory activity is observed with the plant extract and can be linked with the appearance of both the hydroxycinnamic acids and benzophenones [29]. The reduction in nitrite concentration caused by the plant extract suggests the potential of the plant extract in reducing inflammatory cytokines, which can be linked with the effect of benzophenones identified [41]. The lower production of nitrites can also be linked to reduced macrophage infiltration associated with the presence of hydroxycinnamic acids in the plant extract [46]. Besides the inhibitory activity of hydroxycinnamic acids in *G. livingstonei* aqueous leaf extract against macrophage infiltration, a possible mechanism behind the reduction of nitrites could be the increase in anti-inflammatory agents such as adiponectin, caused by hydroxycinnamic acids [46]. The anti-inflammatory effect observed can potentially ameliorate inflammation-induced insulin resistance and subsequently improve glucose metabolism in DM [69]. The synthesis of molecules caused by the reaction of excess glucose with macromolecules leads to the infiltration of immune cells, which subsequently release cytokines that accumulate in organs such as the kidneys and cause apoptosis and tissue damage [9]. The potential reduction of inflammatory cytokines by the leaf extract of *G. livingstonei* can potentially reduce tissue damage associated with DM by reducing apoptosis. In relation to the previously reported delayed wound healing caused by excess inflammatory cytokines, the anti-inflammatory effects of *G. livingstonei* can potentially improve wound healing complications in DM [8]. A similar previous study has also revealed the association of oxidative stress-induced myocardial damage with the excessive release of inflammatory cytokines in DM [70]. The collective anti-inflammatory and antioxidant effects of *G. livingstonei* observed in this current study can potentially treat cardiovascular complications in diabetic individuals.

### 2.6. Glucose Uptake and Utilisation

The rise in blood glucose level is highly dependent on the glucose uptake by cells, normally initiated by the binding of insulin to cell receptors, and the utilisation of the glucose taken in [71]. Besides the absorption of glucose into the blood, glucose uptake from the blood and utilisation by the cells are contributing factors to the balance of blood sugar levels [72]. Although C3A hepatic cells treated with a concentration of 62.5 μg/mL of the plant extract showed a highly significant (*p* < 0.05) decrease in glucose uptake, other concentrations showed a comparable effect in glucose uptake to that of insulin (Figure 6A) It was found that phenolic compounds such as hydroxycinnamic acids present in the extract promote the secretion and action of insulin in cells in vivo [29]. However, the present study was performed in vitro, and it was not possible to observe such effects. A slight, but significant (*p* < 0.05) increase in glucose utilisation was observed with the highest concentration of treatment (125 μg/mL). Glucose utilisation was comparable between the lower concentrations and the control, with no significant (*p* > 0.05) difference observed (Figure 6B). A previous similar study revealed the effect of benzophenones on the increase in fuel utilisation through carbohydrate metabolism by increasing glucose oxidation [40]. This present study supports the previous findings, as the link between the presence of benzophenones and the increase in glucose utilisation is observed [40]. Although the intracellular glucose concentration was not determined, given that glucose uptake through glucose transporter proteins is the rate-limiting step for cellular glucose consumption, it may be assumed that the amount of glucose remaining indirectly reflects glucose uptake. There was no significant (*p* < 0.05) increase in glucose utilisation in cells treated with the plant extract. No cytotoxicity was observed with the treatment of the cells with all the plant extract concentrations, as depicted in Figure 6C. An increase in glucose utilisation can accelerate the clearance of glucose in cells, which can lead to a feedback reaction increasing the production and secretion of insulin.

### 2.7. α-Glucosidase and Lipase Inhibition

Lipase inhibition was used to evaluate the role of the aqueous leaf extract of *G. livingstonei* in the reduction of fat absorption in the intestines and the development of obesity [17]. The leaf extract showed some inhibitory activity against lipase. However, the lipase inhibitory activity was significantly (*p* < 0.05) low at all concentrations of the plant extract compared to the control as depicted in Figure 7A. There was no significant difference in lipase inhibition among the plant extract concentrations. Although the lipase inhibitory activity exhibited by the plant is very low, these findings do not deduce the lower effectiveness of the extract in the management of obesity or dyslipidaemia but rather suggest an alternative additional pathway for the reduction of fats exhibited by the extract constituents. The AMP-activated protein kinase (AMPK) pathway activated by benzophenones identified in the extract can be a possible mechanism by which fat reduction can be achieved through treatment with the extract [40]. It has been concluded from a previous study that benzophenones activate the AMPK signalling pathway, leading to the downregulation of downstream proteins, such as the sterol regulatory element binding protein (SREBP)-1c, and reducing triglyceride accumulation in the cells [41]. It is evident from the findings that *G. livingstonei* exhibits its antidiabetic effect mostly targeting the carbohydrate metabolic enzymes rather than triglycerides. In the process of carbohydrate metabolism (postprandial), the breakdown of carbohydrates is catalysed by carbohydrate enzymes such as α-glucosidase and α-amylase [10]. These enzymes promote the absorption of glucose from the intestines into the blood. Inhibition of α-glucosidase is of importance in glycaemic control for diabetic individuals as it slows down the rise of glucose in the blood, improving hyperglycaemia. Several previous studies have recorded comparable or lower inhibitory activity of other medicinal plants compared to acarbose [1,2]. Interestingly, α-glucosidase inhibitory activity was found to be significantly (*p* < 0.05) higher compared to the standard drug (acarbose) as shown in Figure 7B. The abundance of flavonoids in the leaf extract contributed to the high α-glucosidase inhibitory activity of the extract [27].

## 3. Materials and Methods

### 3.1. Chemicals

All reagents and chemicals used in this study were purchased from Sigma-Aldrich, unless stated otherwise. The following chemicals were used: α-glucosidase (from *Sacchromyces cereviciae*), monobasic sodium phosphate, dibasic sodium phosphate, pNPG, acarbose, MTT, dimethyl sulfoxide (DMSO), minimal essential medium (MEM) and phosphate-buffered saline (PBS) without Ca^2+^ and Mg^2+^ purchased from Cytiva in Marlborough, MA, USA, foetal bovine serum (FBS), non-essential amino acids and penicillin/streptomycin purchased from Biowest in Nuaillè, France, Lipase from porcine pancreas, Tris-HCl, p-nitrophenyl palmitate (pNPP), isopropanol, gum Arabic, sodium deoxycholate, Triton X-100, orlistat, RPMI-1640 Medium, RAW 264.7 mouse macrophages (purchased from Cellonex, Johannesburg, South Africa), Sulfanilamide Solution and NED Solution (purchased from Promega, Madison, WI, USA), Lipopolysaccharide (LPS) and aminoguanidine, RPMI1640 culture medium (from GE Healthcare Life Sciences, Logan, UT, USA), C3A hepatocarcinoma cells (purchased from the ATCC, Manassas, VA, USA).

### 3.2. Plant Material

The leaves of *G. livingstonei* were harvested in 2022 during the summer season (December) from the Brackenridgea Nature Reserve in Thengwe, Limpopo, South Africa. The plant was authenticated by Prof. Tshisikhawe at the University of Venda (voucher number: MNU002/10/22), and a specimen of the plant was deposited at the University of Venda herbarium. The leaves of *G. livingstonei* (100 g) were washed with tap water and dried for 5 days in the shade, after which they were crushed into powder and stored in an airtight container.

### 3.3. Plant Extraction

The powdered leaves of *G. livingstonei* were weighed to up to 100 g and soaked in 1000 mL of hot (100 °C) distilled water overnight (24 h). The liquid was extracted after 24 h with the leaf material removed and discarded. The extract was further filtered using size 2 and 3 paper filters. The filtrate was dried overnight using a freeze dryer. The final dried extract weight was 35.48 g and the percentage yield was 35.48%. The extract was stored at −20 °C in the freezer until use. This explained method of extraction was done following a modified version outlined by Zhao and colleagues [73].

### 3.4. Phytochemical Analysis

#### 3.4.1. Determination of Total Polyphenol

Total polyphenol content was estimated following the methods used by Alabi and colleagues, [74]. A standard curve was generated from the preparation of gallic acid using different concentrations. A volume of 125 μL of Folin reagent (200 mM) was added to 25 μL of the samples (extracts) in 96-well plate wells. After 5 min, 100 μL of sodium carbonate (7.5% *w*/*v*) was added. The plate was incubated for 30 min at 37 °C and absorbances were measured using a Multiskan Spectrum plate reader (Thermo Fisher Scientific, Waltham, MA, USA) at 765 nm. The results were expressed as mg of gallic acid equivalent to a gram of the dry mass of the plant material.

#### 3.4.2. Determination of Flavanol Content

The concentration of flavanol was quantified using the methods followed by Alabi and colleagues [74] and was expressed as mg of quercetin equivalent per gram of the plant (mg QE/g DM). Quercetin was used as a standard. The samples and the standard (12 μL) were added to plate wells of a 96-well plate, followed by 12.5 μL of 0.1% HCl and 225 µL of 2% HCl, and incubated for 30 min at room temperature. The plate was read using a Multiskan Spectrum plate reader (Thermo Fisher Scientific, Waltham, MA, USA) at a wavelength of 360 nm.

#### 3.4.3. Determination of Total Alkaloid

Following the methods used by Alabi and colleagues [74], the total alkaloid content was calculated against a standard curve prepared using a series of dilutions of 0.1 mg/mL atropine in 2 M sodium phosphate buffer (pH 4.7). A mixture of 5 ml of 2 M sodium phosphate buffer (pH 4.7) with 500 mL of the sample/extract and 5 mL of BCG solution was prepared and mixed with 12 mL of chloroform. Two layers were expected to form, and if so, 300 µL of the lower yellow layer was drawn and dispensed into the wells of a 96-well plate. The plate was read at 470 nm using a multiskan spectrum plate reader (Thermo Fisher Scientific, Waltham, MA, USA).

#### 3.4.4. Determination of Flavanol Content

The methods reported by Alabi and colleagues [74] were followed to determine flavanol content. Briefly, 1 mM of catechin will be dissolved in series as a standard. Twenty-five microliters (25 µL) of the samples and the standards were pipetted into the wells of a 96-well plate, and 275 µL of DMACA was added to each well. The plate was incubated for 30 min at room temperature and read at 640 nm. DMACA solution was prepared by adding DMACA to a methanol-HCl mixture at a ratio of 3:1 to a final concentration of 10 µg/mL. The concentration of flavanols was expressed as mg of catechin per gram of the dry mass of the plant material (mg catechin/g DM).

### 3.5. Ultra-High Performance Liquid Chromatography Mass Spectrometry (UHPLC-MS) Analysis

Briefly, phytochemical analysis of *Garcinia livingstonei* extracts was done using a Waters Synapt G2 Quadrupole time-of-flight (QTOF) mass spectrometer (MS) connected to a Waters Acquity ultra-performance liquid chromatograph (UPLC) (Waters, Milford, MA, USA) or high-resolution analysis. Electrospray ionization was used in negative mode with a cone voltage of 15 V, a desolvation temperature of 275 °C, at 650 L/h, and with the rest of the mass spectrometry (MS) settings optimized for best resolution and sensitivity. Data was acquired by scanning from *m*/*z* 150–1500 in resolution mode as well as in MS^E^ mode. Two channels of MS data were captured in the MS^E^ mode, the first at a low collision energy (4 V) and the second at a collision energy ramp (40–100 V), which also allowed for the acquisition of fragmentation data. Leucine enkephalin was used as the reference mass for accurate mass determination, and the instrument was calibrated with sodium formate. Separation was achieved on a Waters T3 HHS, 2.1 mm × 150 mm, 1.7 μm column. The mobile phase consisted of solvents A and B, with the former containing 0.1% formic acid in water and the latter containing 0.1% formic acid in acetonitrile, and the injection volume was 2 μL. A linear flow (flow rate = 0.3 mL/min) of the solvent was maintained for 22 min from a 100% gradient for solvent A (1 min) to 28% for solvent B. The gradient was then increased to 40% in solution B for 50 s after which a wash step took over for 1.5 min at 100% solvent B. At the end of the wash step, re-equilibration occurred for 4 min. The temperature of the equipment was kept at 55 °C.

Data was processed and analysed under MS^E^ acquisition mode. The retention time (RT) ranged from 0.0–29.0 min with a tolerance of ±0.2 min. The mass accuracy tolerance was 5 ± 5 ppm, while mass range was 40–1200 Da. Analysis was performed in negative ion mode and the adducts taken into consideration were, [M-H]^−^, [M-2H]^2−^, [M-H-H_2_O]^−^, [M + Cl]^-^, [M-H-CO_2_]^−^, [M + HCO_2_]^−^, and [2M-H]. Data processing and acquisition were performed using MassLynx^TM^ software version 4.1 (Waters, Milford, MA, USA) and the conversion of the format from project files (.PRO) to NetCDFfiles (.CDF) was done using Databridge in massLynx and MZmine (version 3). *m*/*z* tolerance was set at 0.01 or 10 ppm for processing the Q-ToF data. Compounds were detected according to their RT, MS1, MS2 (*m*/*z*) and UV. During the identification of metabolites, retention times were aligned after the selection of the spectra. The unknown compounds were then detected and grouped, and their compositions were predicted, after which the list was searched, and the background compounds were marked. The N-rule and seven heuristic rules were considered during the prediction of the compounds’ composition to constrain all possible structures. The following N-rule and seven heuristic rules were followed: (1) restrictions for the number of elements, (2) LEWIS and SENIOR chemical rules, (3) isotopic patterns, (4) hydrogen/carbon ratios, (5) element ratio of nitrogen, oxygen, phosphor, and sulphur versus carbon, (6) element ratio probabilities, and (7) presence of trimethylsilylated compounds.

For the assignment of tentative names to the compounds, criteria were set as follows: (1) Accurate mass match: The masses were matched and linked to Metlin (http://metlin.scripps.edu/index.php (accessed on 15 October 2023)), MassBank (http://www.MassBank.jp) (accessed on 15 October 2023)), NIST (http://chemdata.nist.gov/ (accessed on 15 October 2023)), and ReSpect (http://spectra.psc.riken.jp/(accessed on 15 October 2023)). All compounds whose accurate mass error (AME) was > 5 ppm were considered unidentified [75]. (2) Mass fragmentation pattern: In order to identify compounds using their mass fragmentation, retention time, and ionization modes, a few phenolic compounds were used as a standard of comparison and were run under the same conditions. However, not all standards were used in this experiment due to the presence of many compounds identified during UPLC-ESI-QTOF-MS analysis. Therefore, the MS^1^ and MS^2^ fragment ions of previous similar compounds were found in the literature and databases. (3) Carbon atoms: In cases where there were several isotopes of compounds, the calculation of carbon atoms at the peak was used to limit false identifications.

The sensitivity of this method was previously validated [76]. Briefly, chemical markers were set according to the calculated limits of detection (LODs) and limits of quantification (LOQs). Phenolic acids were detected around the UV absorption of 300, 309, and 322 nm. Flavonols were detected at 254, 255, and 354 nm, while flavanols were detected at 278 nm. Different concentrations (3.9; 7.8; 15.6; 62.5; 125.0; and 250.0 mg/L) of the standards (epicatechin and catechin) were injected, and the UV absorptions were obtained to generate a calibration curve. The correlation coefficient of the linear regression model was considered (R > 0.99). The intra- and inter-repeatability of the RT of compounds in the standard ranged from 0.14 to 3.14% and from 1.01 to 2.90%, respectively.

### 3.6. Antioxidant Capacity

#### 3.6.1. Ferric Reducing Antioxidant Power Assay

The ferric reducing antioxidant power assay (FRAP) method, followed by Alabi and colleagues [74], was used to determine the ferric reducing power of the plant extracts. Briefly, the sample and standard (10 µL) were dispensed into the wells of a 96-well plate, and 300 µL of FRAP reagent was added to the wells. The plate was incubated for 30 min at room temperature and read at 593 nm in a spectrophotometer. The FRAP reagent was prepared by mixing 300 mM acetate buffer (pH 3.6)/TPTZ solution, iron (III) chloride hexahydrate solution, and distilled water at a ratio of 10:1:1:2. The standard dilutions were made using ascorbic acid. The results were expressed as mg of ascorbic acid per gram of the plant extract (mg AAE/g sample).

#### 3.6.2. Trolox Equivalent Antioxidant Capacity (TEAC) Assay

The antioxidant activity of the sample was compared to the activity of Trolox as a standard in this study, following the method used by Alabi and colleagues [74]. A serial dilution was prepared from a Trolox stock solution of 1000 mg/L. A volume of 275 µL of 2,2-azino-bis-3-ethylbenzothiazolinee-6-sulphonic acid (ABTS) (0.4 mg/mL) was added to 25 µL of the samples in a 96-well plate. The plate was incubated at room temperature for 30 min to allow the reaction to occur. Subsequently, the plate was read at 593 nm in a Multiskan Spectrum plate reader (Thermo Fisher Scientific, Waltham, MA, USA).

#### 3.6.3. 2,2-diphenyl-1-picrylhydrazyl (DPPH) Assay

The DPPH value of the plant extract was determined using Trolox as a standard of comparison. Trolox standard diluted solutions were prepared as a 1000 mg/L stock solution of Trolox. A volume of 25 µL of the extract sample and the standard was plated in a 96-well plate, after which 275 µL of DPPH with a concentration of 0.4 mg/mL was added. The plate was left for 30 min at room temperature, then read at 593 nm in a Multiskan Spectrum plate reader (Thermo Fisher Scientific, Waltham, MA, USA) [74].

### 3.7. Anti-Inflammatory Effect Determination

The mouse macrophage cell line, RAW 264.7, was used to determine anti-inflammatory activity. RAW 264.7 cells were seeded in RPMI1640 culture medium supplemented with 10% FBS (RPMI complete medium) into 96-well plates at a density of 1 × 10^5^ cells per well and allowed to attach overnight. The following day, the spent culture medium was removed and 50 µL of sample aliquots (diluted in RPMI complete medium) added to give final concentrations of 50, 100, and 200 µg/mL. To assess the anti-inflammatory activity, 50 μL of lipopolysaccharide (LPS) (final concentration of 1 µg/mL)-containing medium was added to the corresponding wells. Aminoguanidine (AG) was used as the positive control at 100 μM. Cells were incubated for a further 24 h. To quantify NO production, nitrite was measured in spent culture medium using the Griess reagent system based on the diazotization reaction originally described by Griess (1879). The spent culture medium was transferred (50 μL) to a new 96-well plate, and 50 μL of sulphanilamide solution was added. Plates were incubated for 10 min at room temperature in the dark, then 50 μL of NED solution was added, followed by incubation for 10 min at room temperature in the dark. Absorbance was measured at 540 nm using a BioTek^®^ PowerWave XS spectrophotometer (Winooski, VT, USA). A standard curve using sodium nitrite was used to determine the concentration of NO (µM) in each sample. To confirm the absence of toxicity as a contributory factor, cell viability was assessed using MTT as described under glucose utilisation and uptake [77].

### 3.8. Glucose Uptake and Utilisation

#### 3.8.1. Cell Line Maintenance

Glucose uptake and utilization were tested on C3A hepatocarcinoma cells, which were maintained in 10 cm culture dishes in complete medium (MEM with 1% NEAA, 10% FBS, and 1% Pen-Strep) and incubated at 37 °C in a humidified atmosphere with 5% CO_2_.

#### 3.8.2. Glucose Uptake

Cells were seeded in 96 well plates (2 × 10^4^ cells/well, 100 μL aliquots) and left overnight to attach. Different concentrations of the treatment (extract) were prepared in complete medium (31.25, 62.5, and 125 μg/mL) and added to cells and incubated for 24 h. Treatment was removed after 24 h of incubation, and cells were transferred to new 96-well plates, after which 200 μL glucose oxidase (prepared from the mixture of 1 mU/mL *Aspergillus niger* glucose oxidase with 0.4 mM 4-aminoantipyrine, 2.5 U/mL horseradish peroxidase, 3 mM phenol, 0.25 mM EDTA, and 0.5 M PBS (pH 7.0)) was added. The plate was kept at room temperature for 15 min and read at 510 nm using a BioTek^®^ PowerWave XS spectrophotometer (Winooski, VT, USA). The glucose standards in the wells contained culture mediums and the incubation buffer, with no cells. The mean difference (in mM) of the concentrations of glucose remaining after the reaction, between glucose standards and the samples was used to determine glucose intake [77].

#### 3.8.3. Glucose Utilisation

Cells were seeded in 96 well plates (2 × 10^4^ cells/well, 100 μL aliquots) and left overnight to attach. Different concentrations of the treatment (extract) were prepared in complete medium (31.25, 62.5, and 125 μg/mL) and added to cells and incubated for 24 h. Treatment was removed after 24 h of incubation, and cells were transferred to new 96 well plates. The remaining medium was aspirated, and cells were washed with 100 μL PBS and 25 μL of incubation buffer (RPMI-1640 diluted with PBS containing 0.1% BSA to a final glucose concentration of 8 mM) was added to the cells, followed by 4 h of incubation. A volume of 200 μL glucose oxidase reagent was added to the plates, followed by incubation for 15 min at room temperature. Absorbance was measured at 510 nm using a BioTek^®^ PowerWave XS spectrophotometer (Winooski, VT, USA). Cell-free wells containing incubation buffer and complete culture mediums were used as glucose standards. Glucose uptake was determined as a function of the concentration of glucose (mM) remaining and expressed as the difference between the mean of the standard and test samples [77].

### 3.9. Cytotoxicity Determination-MTT Assay

The MTT assay was performed on the cells tested for glucose utilisation and uptake to ensure the viability of the cells during the glucose uptake and utilisation assays. After the assays, the remaining treatment medium was aspirated from all wells after 24 h and 100 μL of complete medium containing 0.5 mg/mL MTT was added to the cells. The cells were incubated for 1 h at 37 °C. MTT was removed, and 100 μL DMSO was added to each well to solubilise the formazan crystals. Absorbance was read at 540 nm using a BioTek^®^ PowerWave XS spectrophotometer (Winooski, VT, USA) [77].

### 3.10. Lipase Inhibition

Samples were diluted in assay buffer (100 mM Tris-HCl, pH 8.0) to concentrations of 500, 250, 125, 62.5, and 31.25 μg/mL. The pancreatic lipase inhibition assay was performed as described by Pringle and colleagues [78]. A volume of 40 μL of the lipase enzyme was added to 10 μL of the sample in a 96-well plate in quadruplicate. The plate was incubated at 37 °C for 15 min, after which 170 μL of substrate with reaction buffer was added. The plate was incubated again at 37 °C for 25 min. Absorbance was measured at 405 nm using a BioTek^®^ PowerWave XS spectrophotometer (Winooski, VT, USA). No enzyme and no substrate controls were included, and the percentage lipase inhibition was calculated using:$$\%Lipase\;inhibition=\frac{A405\;nm\;of\;blank-A405\;nm\;of\;test\;sample)}{A405\;nm\;of\;blank}\times 100$$

### 3.11. α-Glucosidase Inhibition

The inhibitory capacity of the extract was determined and compared to that of acarbose, a standard drug. Different comparable concentrations of the plant extract and acarbose were prepared (15.625, 31.25, 62.5, 125, and 200 ug/mL) using distilled water. Briefly, 100 μL of α-glucosidase was added to 50 μL of each well of the concentration of samples and the standard drug (with phosphate buffer as a negative control) in a 96-well plate and incubated at room temperature for 10 min. A substrate buffer solution (50 μL), prepared from 20 mM phosphate buffer, pH 6.8, and 5 mM of pNPG was then added into the wells to start the reaction. The 96-well plate was incubated for 30 min at 35.5 °C. During the reaction, the colour of the solutions turned yellow due to the product of the breakdown of pNPG by the α-glucosidase enzyme. The %inhibition was observed by the shade of yellow, with the darker shades representing lower %inhibition. The %inhibition of each extract and acarbose concentration was calculated using the formulas below [79].
$$\%\;\alpha -glucosidasee\;inhibition\;=\;\frac{A405\;nm\;of\;blank-A405\;nm\;of\;test\;sample)}{A405\;nm\;of\;blank}\times 100$$
%inhibition= [(Ac − As)/Ac] × 100

Ac = Absorbance of the negative control

As = Absorbance of the sample

### 3.12. Statistical Analysis

Statistical analysis was performed using GraphPad prism version 5. Analysis was done using One-way ANOVA (non-parametric) with Tukey post-test. Significance is denoted by letters on bars where bars with different letters show significant differences (*p* < 0.05), and bars with the same letter have no significant difference (*p* > 0.05).

## 4. Conclusions

The findings of this study revealed the therapeutic potential of *G. livingstonei* for the treatment of DM. The identification of the major antioxidant and anti-inflammatory compounds shows the potency of the plant extract in the amelioration of oxidative stress and inflammation in diabetic individuals. It is also evident that the plant can potentially improve both carbohydrate and lipid metabolism, which can promote the management of obesity and hyperglycaemia. Although the preparation of the extract with a different solvent, such as ethanol, and the extraction period could play a role in the determination of the effects of *G. livingstonei* and the quantification of the phytochemical content, this study focused on the traditional method of preparation. Additionally, the use of water in the preparation of extracts has been proven effective and less costly. Comparison and relation to previous studies were insufficient, as the studied plant has limited information in the literature. Due to budget constraints, complementary experimental works, such as the use of several extraction methods and the evaluation of more carbohydrate enzymes, could not be conducted. However, further studies with the use of different solvents and a longer extraction period could be highly useful in the findings. Further isolation of the identified compounds can also be useful in further investigations. Further in vitro and in vivo animal studies of *G. livingstonei* are needed to further validate the results from this study.

## Figures and Tables

**Figure 1 ijms-25-03184-f001:**
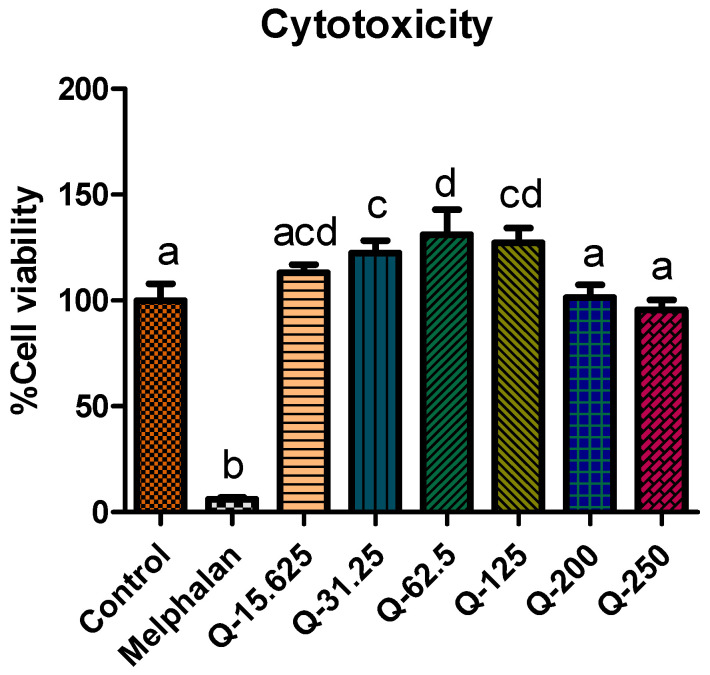
Cytotoxicity of the different concentrations of *G. livingstonei* leaf extract in μg/mL against C3A cells. Cells were treated for 48 h. Error bars indicate the standard deviation of quadruplicate values performed as a single experiment. Melphalan (10 μM) was used as a positive control. The bar graph represents the mean of quadruplicate values. Error bars represent the standard deviation of the mean. Statistics were performed using one-way ANOVA and Tukey’s post-test. Letters on the bars show significant differences, where bars with different letters are significantly different (*p* < 0.05).

**Figure 2 ijms-25-03184-f002:**
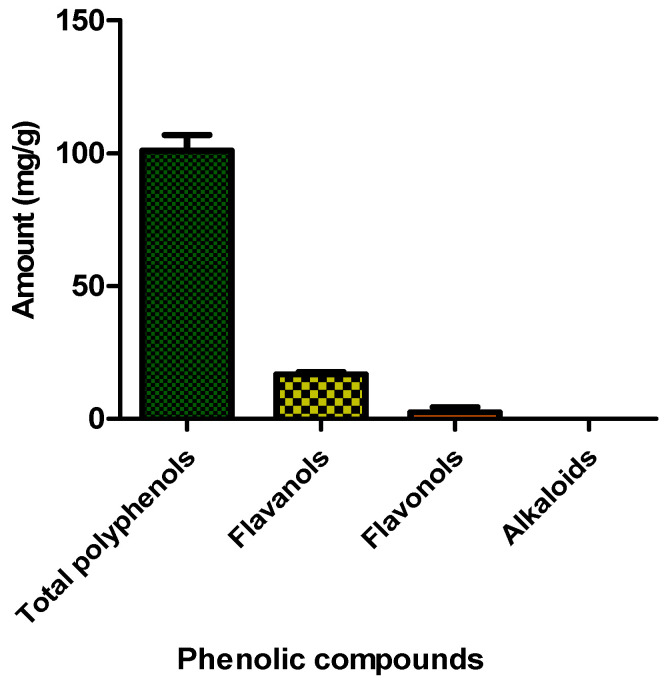
Quantification of phytochemical compounds in *G. livingstonei* leaf extract. Error bars indicate the standard deviation of the mean of the quadruplicate values.

**Figure 3 ijms-25-03184-f003:**
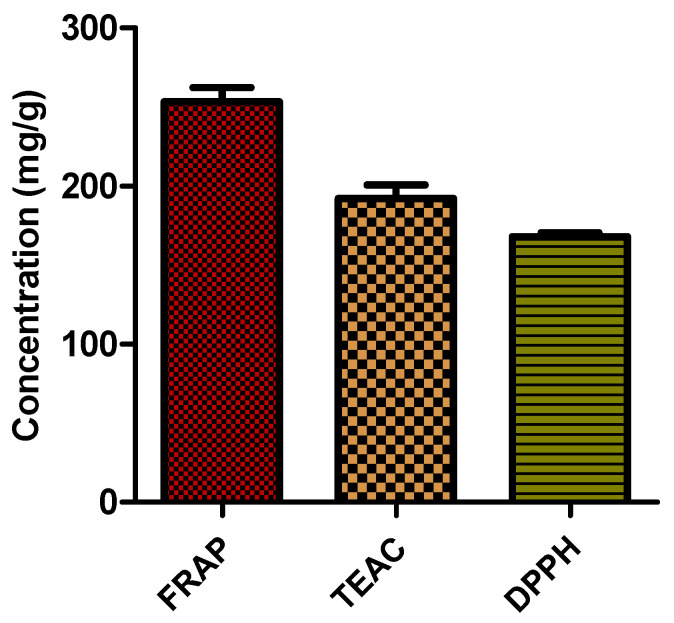
Antioxidant capacity of *G. livingstonei* leaf extract. Values of ferric reducing antioxidant power (FRAP), Trolox equivalent antioxidant capacity (TEAC), and 2,2-diphenyl-1-pycrylhydrazyl (DPPH) were determined. Error bars indicate the standard deviation of the mean of the quadruplicate values.

**Figure 4 ijms-25-03184-f004:**
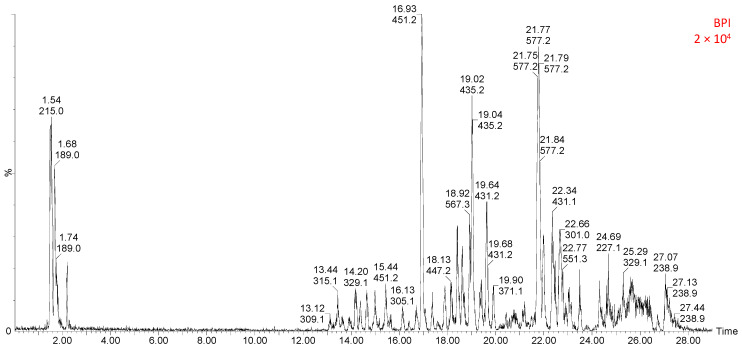
UHPLC-ESI-MS base peak chromatogram for the extract of *Garcinia livingstonei* analysed in the ESI negative mode.

**Figure 5 ijms-25-03184-f005:**
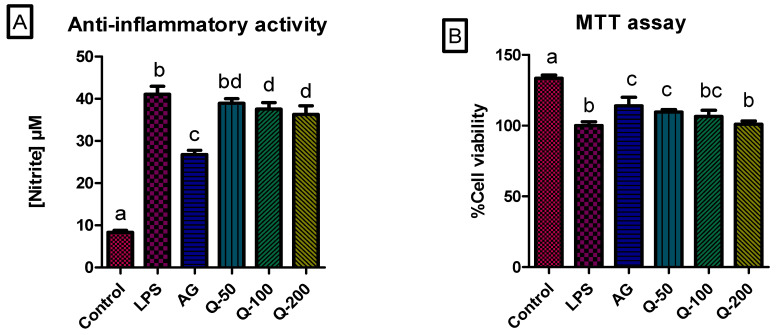
(**A**) Nitric oxide production in LPS-activated macrophages treated with samples. The bar graph represents the quadruplicate values of one experiment. Error bars represent the standard deviation of the mean. (**B**) Cell viability (%) of LPS-activated macrophages after 24 h of exposure to treatments. The bar graph represents the mean of the quadruplicate values. Error bars represent the standard deviation of the mean. Statistics were performed using one-way ANOVA and Tukey’s post-test. Letters on the bars show significant differences, where bars with different letters are significantly different (*p* < 0.05).

**Figure 6 ijms-25-03184-f006:**
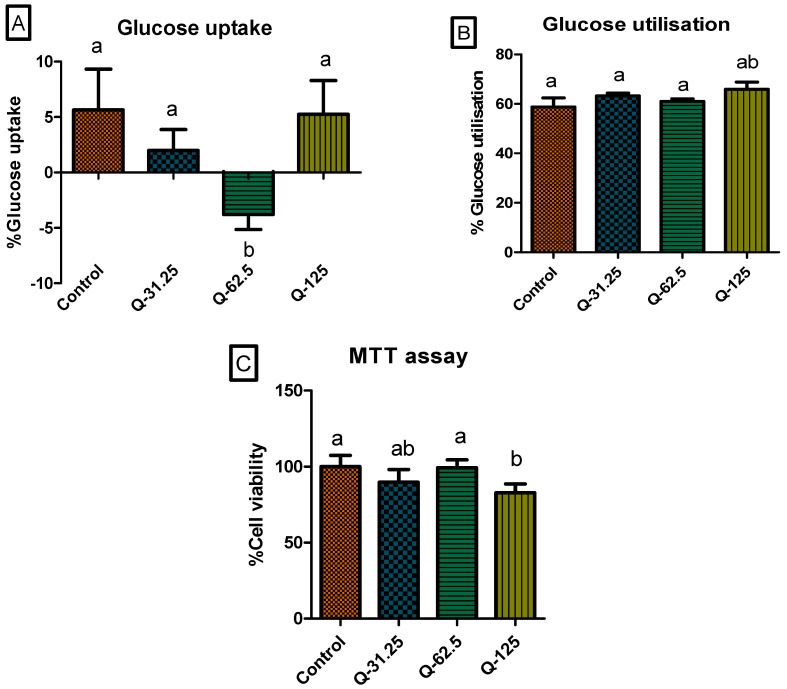
Glucose utilization (%) after 24 h of treatment with different concentrations of the extract (31.25, 62.5, 125 μg/mL) in C3A cells (**A**) and glucose uptake (%) after 4 h in C3A cells, following 24 h of the pre-treatment (**B**). Results were normalized to cell viability as determined using the MTT assay (**C**). The bar graph represents the mean of the triplicate values. Error bars represent the standard deviation of the mean. Statistics were performed using one-way ANOVA and Tukey’s post-test. Letters on the bars show significant differences, where bars with different letters are significantly different (*p* < 0.05).

**Figure 7 ijms-25-03184-f007:**
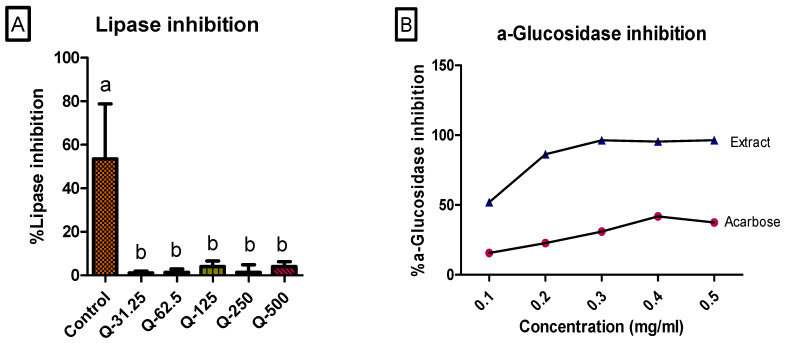
Pancreatic lipase inhibition and α-glucosidase inhibition. (**A**) 100 µM Orlistat was used as a positive control. Error bars indicate the standard deviation of the mean of four replicates. The bar graph represents the quadruplicate values of one experiment. Error bars represent the standard deviation of the mean. Statistics were performed using one-way ANOVA and Tukey’s post-test. Letters on the bars show significant differences, where bars with different letters are significantly different (*p* < 0.05). (**B**) α-glucosidase inhibition was compared to acarbose as a positive control.

**Table 1 ijms-25-03184-t001:** Phytochemicals screened from the extract of *Garcinia livingstonei* using UHP-LCMS.

No	t_R_ (min)	UV λmax (nm)	*m*/*z* [M-H]^−^	MS/MS	Tentative Name	References
1	1.54		215.0224/225.0520	179, 165, 133	Dimethoxycinnamic acid monohydrate	[47]
2	1.68		188.9939	127	2,6-Diaminopimelic acid	[48]
3	2.21	262, 290	191.0101	179, 119, 149	p-Coumaric acid ethyl ester	[49]
4	14.20	254, 290	329.0814	167, 279	Vanilic acid hexoside	[50]
5	14.38	286, 310	423.0862	315, 291	Unidentified	-
6	14.65	278, 311	469.1545	423, 391, 343, 203	Unidentified	-
7	14.98	256, 301	361.0706	272, 193	2,4,6-Trihydroxy-2^1^,5^1^-dimethoxybenzophenone derivative	[38]
8	15.44	236, 314	451.2101	351, 272, 361	3^1^-β-Glucosyloxy-4,4^1^-dihydroxy-2,6-dimethoxy-benzophenone	[38]
9	16.13	271	305.0615	236, 162	(±)-Gallocatechin	[50]
10	16.72	285 sh	583.2595	451, 293	3-hydroxyphloretin 2′-O-xylosylglucoside	[51]
11	16.93	281	451.2140	387, 441, 113, 45	Epicatechin-3-O- hexoside	[52]
12	17.36	264	417.0993	159	Kaempferol-3-O-pentoside	[53]
13	17.89	280	577.1384	449	Procyanidin B2	[54]
14	18.17	280	447.1834	407, 437, 261, 327	Kaempferol-3-O-hexoside	[55]
15	18.42	300	473.1236	439, 215	3-O-malonyl-5-O-(E)-caffeoylquinic acid derivative	[42]
16	18.51	281	439.1740	239	3-O-malonyl-5-O-(E)-caffeoylquinic acid	[42]
17	18.63	252, 289	431.1884	289, 421	Epicatechin-3-O-xylosyl derivative	-
18	18.71	279	289.0652	271, 215	Epicatechin	[56]
19	18.92	280	567.2673	431, 327, 521	Epicatechin -3-O-(4^1^-xylosyl) deohexoside	New
20	19.02	281	435.2204	289, 425	Catechin-3-rhamnoside	[57]
21	19.42	267, 285, 337	435.2199	289, 431, 421	Epicatechin-O-rhamnoside	[57]
22	19.64	285	431.1862	385	Hexoside of kaempferol-3-O-(deoxy)	[52,58]
23	19.90	281	371.0909/441.1890	247, 309, 157	(−)-Epicatechin-3-Ogallate	[59]
24	21.77	270, 344	577.1569	Un fragmented	Procyanidin B1 dimer	[60]
25	21.84	268, 338	577.1582	413, 293, 311	Procyanidin B11 dimer	[61]
26	22.00	268, 335	431.0912	Un fragmented	Epicatechin-3-O- xylosyl derivative	-
27	22.34	270, 335	431.0898	289, 317	Epicatechin-3-O- xylosyl derivative	-
28	22.47	270, 334	316.9897	237, 207, 283	Isorhamnetin	[62]
29	22.68	292	300.9943/563.2272	221, 157	Apigenin-C-pentoside-C-hexoside	[57,58]
30	22.80		551.2685	447, 301, 541, 245, 505	O-Methyl-gallocatechin-3-O-gallate-O-sulfates	[63]
31	22.92		551.2707	301, 447, 401, 541, 319, 169	O-Methyl-(epi)gallocatechin-3-O-gallate-O-sulfate	[63]
32	23.49		553.2888 ^[M-2Na]−^	419, 507	Quercetin-3-O-acetyl-hexoside	[58]
33	24.31	279	187.0867	169, 127	Gallic acid	[56,64]
34	24.69		227.1176		Not identified	
35	25.29		329.0599	150, 145	Fagomine derivative	[65]
36	25.62	302	149.9842		Fagomine	[65,66]
37	27.07		238.8820		Unidentified	

## Data Availability

The data presented in this study are available on request from the corresponding author.

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
