# Peer review of "Antioxidant, Anti-Diabetic, and Anti-Inflammation Activity of Garcinia livingstonei Aqueous Leaf Extract: A Preliminary Study"

_ijms, 2024, doi:10.3390/ijms25063184_

Round 1

Reviewer 1 Report

Comments and Suggestions for Authors

I read with great interest the paper “Preliminary determination of the phenolic compounds profile and antioxidative, antidiabetic, and anti-inflammatory activity effects of aqueous leaf extract of Garcinia livingstonei. by Nethengwe et al.

The content is presented in a logical manner. The English is generally well structured. However, for clarity and flow, you might consider breaking down the lengthy sentences into smaller ones.

Here are some suggestions for improvement:

1.      You might want to modify the title "Preliminary Evaluation of Phenolic Compounds Profile and Assessment of Antioxidative, Antidiabetic, and Anti-Inflammatory Activities of Aqueous Leaf Extract from Garcinia livingstonei”.

2.      Please check font and character dimension.

3.      Introduction: line 58-63. Convensional treatment sure include metformin (even as first line therapy), though the role of glibenclamide and acarbose is really limited (third line therapy). Improve this part.

4.      References in the text should be reported as number ([1]). Moreover, reference style is not appropriate for this journal.

5.      Discuss how antinflammatory and antioxidant effect of glarcinia might impact diabetes (Curr Issues Mol Biol 2023 Aug 12;45(8):6651-6666. doi: 10.3390/cimb45080420.).

6.      Provide more details on the modified Zhao and colleagues' method for phytochemical analysis.

7.      Ensure consistent formatting (e.g., a numbering error in "3.4.5. Determination of flavanol content").

8.      Offer more information about the specific compounds identified, especially those relevant to potential therapeutic effects.

9.      Include information about the sensitivity and specificity of the UHP-LCMS method.

1.   Provide additional information on the antioxidant capacity results.

1.   Include values or ranges for the antioxidant capacity assessments (FRAP, TEAC, DPPH).

1.   Include details on the concentrations used in the anti-inflammatory assay.

1.   Conclusions: Discuss limitations and suggest areas for future research.

Comments on the Quality of English Language

Minor English revision

Reviewer 2 Report

Comments and Suggestions for Authors

Dear Editor and Authors,

The manuscript ' Preliminary determination of the phenolic compounds profile and antioxidative, antidiabetic, and anti-inflammatory activity effects of aqueous leaf extract of Garcinia livingstonei ‘ by Murendeni Nethengwe, Nasifu Kerebba, Kunle Okaiyeto, Chinyerum S. Opuwari and Oluwafemi O. Oguntibeju is a research manuscript on cytotoxicity, phytochemical analysis, antioxidant capacity, anti-inflammatory and antidiabetic effects in aqueous extract of Garcinia, livingstonei leaves. There is few research on Garcinia, livingstonei leaves composition and properties. The topic is interesting but…

The Authors themselves soon in the title suggest that the phenolic compounds profile is a preliminary and I fully agree with them. More research on Garcinia livingstonei polyphenols is for sure needed. I have never met epicatechin-O-rhamnoside and catechin-3-rhamoside before. Flava-3-ols bids together or to gallates not with sugars. New knowlegde on Garcinia, livingstonei leaves but insufficient. The Authors have found compounds that do not exist like glycosides of flavan-3-ols.

No standards were used for identification, even the most basic like quercetin. Improve significantly the identification of polyphenols done by UHPLC-MS to not show substances that do not exist in nature. No proper identification of compounds decreases the value of the work performed. Without proper identification, the manuscript cannot be improved. Figures are of good technical quality but the identification of polyphenols is not done properly.

Abstract

UHPLC-MS is usually the abbreviation

Line 379 spectrophotometer details should be given (name, producer, country)

Yours sincerely,

Comments on the Quality of English Language

Dear Editor and Authors,

The quality of English in the manucript is proper,

Yours sincerely,

Round 2

Reviewer 1 Report

Comments and Suggestions for Authors

No further comments

Reviewer 2 Report

Comments and Suggestions for Authors

Dear Editor and Authors,

The manuscript is significantly improved after the revision. But, still the Table 1 needs some improvement. You cannot say that all the compounds are new. There is plenty of literature on identification of polyphenols like gallic acid, isorhamnetin ect. Do the UV Max and m/z {M-H]- match with the ones in other raports? Even in a different plant. It may be first identified in your plant but it already has benn identified previuosly,

Cordially

Round 3

Reviewer 2 Report

Comments and Suggestions for Authors

Dear Editor and Authors,

The manuscript after the revision is significantly improved,

Cordially